# Combined Transcriptome and Proteome Analysis Provides Insights into Petaloidy in Pomegranate

**DOI:** 10.3390/plants12132402

**Published:** 2023-06-21

**Authors:** Yan Huo, Han Yang, Wenjie Ding, Tao Huang, Zhaohe Yuan, Zunling Zhu

**Affiliations:** 1College of Landscape Architecture, Nanjing Forestry University, Nanjing 210037, China; yanhuo@njfu.edu.cn (Y.H.);; 2Southern Modern Forestry Collaborative Innovation Center, Nanjing Forestry University, Nanjing 210037, China; 3Jinpu Research Institute, Nanjing Forestry University, Nanjing 210037, China; 4Research Center for Digital Innovation Design, Nanjing Forestry University, Nanjing 210037, China; 5College of Traditional Chinese Medicine, Weifang Medical University, Weifang 261053, China; 6College of Landscape Engineering, Suzhou Polytechnic Institute of Agriculture, Suzhou 215008, China; 7College of Forestry, Nanjing Forestry University, Nanjing 210037, China; 8College of Art and Design, Nanjing Forestry University, Nanjing 210037, China

**Keywords:** *Punica granatum*, petaloidy, association analysis, transcriptome, proteome

## Abstract

Petaloidy leads to a plump floral pattern and increases the landscape value of ornamental pomegranates; however, research on the mechanism of petaloidy in ornamental pomegranates is limited. In this study, we aimed to screen candidate genes related to petaloidy. We performed transcriptomic and proteomic sequencing of the stamens and petals of single-petal and double-petal flowers of ornamental pomegranates. Briefly, 24,567 genes and 5865 proteins were identified, of which 5721 genes were quantified at both transcriptional and translational levels. In the petal and stamen comparison groups, the association between differentially abundant proteins (DAPs) and differentially expressed genes (DEGs) was higher than that between all genes and all proteins, indicating that petaloidy impacts the correlation between genes and proteins. The enrichment results of transcriptome, proteome, and correlation analyses showed that cell wall metabolism, jasmonic acid signal transduction, redox balance, and transmembrane transport affected petaloidy. Nine hormone-related DEGs/DAPs were selected, among which ARF, ILR1, LAX2, and JAR1 may promote petal doubling. Sixteen transcription factor DEGs/DAPs were selected, among which EREBP, LOB, MEF2, MYB, C3H, and trihelix may promote petal doubling. Our results provide transcriptomic and proteomic data on the formation mechanism of petaloidy and a theoretical basis for breeding new ornamental pomegranate varieties.

## 1. Introduction

Increasing the number of petals makes flowers plumper and improves their ornamental value. Therefore, petaloidy is a popular topic in flower pattern research [1,2]. Currently, research on the molecular mechanisms of petaloidy has mainly been focused on the early stages of the development of floral organs. Researchers have constructed the ABCDE flower development model [3,4,5,6,7,8] and verified the function of homeotic genes in regulating stamen petaloidy in plants, such as the *Petunia hybrid* [9], *Gentiana scabra* [10], and *Camellia japonica* [11]. However, flower development models do not sufficiently explain petaloidy in some plants [12,13,14]. Previous studies have found that hormone-related genes interact with flower development model genes, jointly affecting the development of stamens and petals. For example, the jasmonate synthase gene *DAD1* and the homologous gene *AG* jointly regulate the stamen development of *Arabidopsis thaliana* [15], the auxin response factor gene *ARF* regulates stamen petaloidy in *Rosa hybrida* by inhibiting *AG* transcription [16], and cytokinin and auxin regulate bud induction and organ determination in *Hyacinth orientalis* flowers by homologous gene *HAG1* [17]. The same phenomenon occurs in plants such as *Nelumbo nucifera* [18], *Malus* spp. [19], and *Lagerstroemia speciosa* [20].

Ornamental pomegranate (*Punica granatum*), a species of *Punica* in the Lythraceae family, is widely planted in courtyards, roads, and parks because of its beautiful flowers and fruits. Petaloidy mainly occurs during the flower bud and flowering periods, including stamen petaloidy and transitional petal formation and growth. Transcriptomics and proteomics are the studies of total RNA and total proteins, respectively, in cells or tissues under certain conditions [21,22,23,24]. Transcriptome and proteome association analyses can systematically analyze data such as gene function, expression levels, and related metabolic pathways at the level of transcription and translation, which are obtained using high-throughput sequencing technology [25,26,27]. It has been reported that the petaloidy mechanism was studied during the flowering period using multi-omics. For example, Huo [28] identified genomic variation between three single-petal varieties and three double-petal varieties of pomegranate via whole genome re-sequencing. Lin et al. [18] studied the molecular mechanism of carpel petaloidy in *N. nucifera* through transcriptome and proteome association analysis, and Huang et al. [29] studied the molecular mechanism of lip-like petal and sepal development in *Phalaenopsis* using genomics and transcriptome association analysis. However, there are no reports on petaloidy molecular mechanisms during the flowering period of ornamental pomegranate based on transcriptomics and proteomics.

In this study, we aimed to (i) screen the candidate genes involved in the transformation from stamen to petal stamen and from transition petals to normal petals in ornamental pomegranate by using transcriptome and proteome analyses and (ii) elucidate the molecular mechanism of petaloidy and lay a foundation for cultivating new varieties of ornamental pomegranate.

## 2. Results

### 2.1. Petaloid Phenotype of Ornamental Pomegranate

When compared with the single-petal flower, the double-petal flower of the ornamental pomegranate had a larger flower diameter, more petals, and a different flower shape (Figure 1a,b). The morphological differences between double and single-petal flowers were mainly reflected in the petals and stamens. The stamens of the single-petal flowers developed normally and formed anthers and pollen (Figure 1e). The stamens of the double petals developed abnormally and were constantly petalized during the flower opening process (Figure 1f). The inner petals of the double-petal flowers were mainly composed of petaloid stamens, which formed transitional petals (Figure 1g), while the petals of the single-petal flowers developed normally without transitional petals (Figure 1h). Figure 1c shows that the petaloid stamen number in the double-petal flowers (180.56) was significantly larger than that in the single-petal flowers (0.00), while Figure 1d shows that the petal number in the double-petal flowers (73.00) was significantly larger than that in the single-petal flowers (6.11). The petal number and the petalized stamen number were important morphological indicators that distinguished between the single- and double-petal flowers of pomegranate. The comparison between the stamens of the single-petal flowers (StSi) and stamens of the double-petal flowers including petalized stamens (StDo) and between the petals of the single-petal flowers (PeSi) and petals of the double-petal flowers including transitional petals (PeDo) showed the key dynamic stage of the transformation from the stamens to petaloid stamens and from the transitional petals to normal petals.

### 2.2. Basic Information on Transcriptome Sequencing

The 12 libraries resulted in 78.19 Gb of clean data and 75.68 Gb of high-quality reads (Q > 20). Each sample having more than 6.31 Gb of clean data was retrained and used for further analysis (Table 1). Approximately 91.63–94.37% of the clean short reads were aligned to the reference genome of pomegranate ‘Dabenzi’. The results of the transcript random, transcript coverage, and transcript saturation distribution tests showed that the quality of each sample was good, and there were sufficient sequencing data (Appendix A). A total of 24,567 genes were detected, including 2896 new genes. Additionally, 20,409, 21,175, 18,742, and 20,983 quantifiable genes were identified in PeSi, StSi, PeDo, and StDo, respectively (Appendix A). We found that 32.38–37.66% of genes were expressed in the 1–10 FPKM range and 36.45–46.68% in the range of 10 FPKM (Figure 2a). The correlation analysis showed high reproducibility among the duplicate biological samples (Figure 2b).

### 2.3. Differentially Expressed Genes among the Floral Organs

In total, 10,373 and 5331 differentially expressed genes (DEGs) were identified in the PeSi vs. PeDo and StSi vs. StDo groups, respectively, of which 3207 were common DEGs (Figure 3). Briefly, 1402 genes were upregulated and 8971 genes were downregulated in the PeSi vs. PeDo group and 3389 genes were upregulated and 1942 genes were downregulated in the StSi vs. StDo group (Figure 4). According to the Gene Ontology (GO) functional enrichment analysis (Appendix A and Figure 5, Figure 6 and Figure 7), we found similarities between the PeSi vs. PeDo, StSi vs. StDo, and common DEGs. The similarities were as follows: in terms of the cellular components, the DEGs in the three groups were enriched in the nucleus; in terms of the molecular function, the DEGs in the PeSi vs. PeDo and common DEGs were mainly enriched in DNA helicase and RNA and nucleotide binding. The DEGs in the StSi vs. StDo and common DEGs were mainly enriched in transmembrane transport, oxidoreductase activity, and coenzyme binding; in terms of the biological processes, three groups of DEGs were enriched in DNA repair after damage. In addition, the Kyoto Encyclopedia of Genes and Genomes (KEGG) enrichment analysis revealed that the common DEGs were mainly enriched in the degradation of amino acids and fatty acids, protein transport, and signal transduction pathways (Figure 8). In summary, the DEGs that were involved in stamen petaloidy and transitional petal development were mainly enriched in signal transduction, DNA repair, redox balance, transmembrane transport, translational release, and lignin metabolism (Table 2).

### 2.4. Hormone-Related and Transcription Factor Differentially Expressed Genes Involved in Petaloidy

Previous studies showed that plant hormones and transcription factors (TFs) are involved in stamen petaloidy and petal variation [13]. In this study, the screening of 3207 DEGs resulted in the identification of four hormone-related DEGs, including three auxin-responsive factors (*ARF*s) and one indole-3-acetic acid-amido synthetase (*GH3.17*). Figure 9 shows that the expression level of *ARF*s was highest in PeSi and lowest in StSi, while that of *GH3.17* was highest in StSi, indicating that *ARF*s and *GH3.17* may have a reverse regulatory effect on the floral organ development of pomegranate. Fourteen TF DEGs belonged to the APETALA2-ethylene-responsive element binding protein (EREBP), C3H, LIM, LOB, MADS, MYB, trihelix, and WRKY families. The TF DEGs had different spatial expression patterns (Figure 10).

### 2.5. RNA Sequencing Data Validation through Quantitative Real-Time Polymerase Chain Reaction

To validate the reliability of the RNA sequencing (RNA-Seq) data, 12 genes were chosen at random for quantitative real-time polymerase chain reaction (qRT-PCR) analysis. The correlation coefficient between the qRT-PCR data and transcriptome data fell between 0.815 and 0.996, which is indicative of the high reliability of RNA-seq data (Appendix A).

### 2.6. Basic Information of the Proteome Sequencing

After protein extraction, 2,178,185 secondary spectra were obtained, of which 411,677 were available and the utilization rate was 18.9%. In total, 49,696 peptides were identified through spectral analysis, including 43,127 unique peptides. The peptide length distribution diagram showed that most of the peptides were distributed in the range of 1–20 amino acids, which conformed to the general rule of the trypsin enzyme and higher energy collisional dissociation (Appendix A). A total of 5865 proteins were identified, of which 5727 were quantifiable (Appendix A). The vast majority of the proteins (99%) had molecular weights greater than 10 kDa, with a good distribution range (Figure 11a). The distribution of the protein molecular weights showed that most of the proteins had high coverage (Appendix A). A total of 4791, 5269, 5055, and 4835 quantifiable proteins were identified in the StSi, StDo, PeDo, and PeSi clusters, respectively (Figure 11b). Principal component analysis (Figure 11c) and Pearson’s correlation analysis (Figure 11d) showed that for each of the three floral organs sampled, there was good reproducibility of the data across the three biological replicates. In addition, Figure 11d shows that, among each group of samples, the correlation coefficient between PeSi and PeDo was the highest (0.96), followed by StSi and StDo (0.91), indicating similar expression patterns between these two groups of samples.

### 2.7. Differentially Abundant Proteins Involved in Petaloidy

In total, 419 and 642 DAPs were identified in PeSi vs. PeDo and StSi vs. StDo, respectively, of which 101 were common DAPs (Figure 12). A total of 227 proteins were upregulated and 192 were downregulated in PeSi vs. PeDo. Moreover, 249 proteins were upregulated and 393 were downregulated in StSi vs. StDo (Figure 13). The GO enrichment analysis revealed that the common DAPs were mainly enriched in the cell wall and mitochondrial membrane in terms of the cell composition, various enzymes in terms of the molecular function, and the polysaccharide metabolism and jasmonate signal transduction pathway in terms of the biological processes (Appendix A).

Three hormone-related DAPs were screened from the common DAPs: the IAA-amino acid hydrolase protein (ILR1), jasmonic acid-amido synthetase protein (JAR1), and abscisic acid stress ripening-induced protein (ASR). Among these, the expression of ILR1 and JAR1 in PeDo and StDo was higher than that in PeSi and StSi, whereas the expression pattern of ASR was the opposite (Figure 14). Two TF DAPs belonging to the C3H and trihelix families were obtained, and their expression levels in StSi, StDo, and PeDo increased.

### 2.8. Association Analysis of the Transcriptome and Proteome

#### 2.8.1. Quantitative Relationship between the Transcriptome and Proteome

In total, we identified 24,567 genes and 5865 proteins in PeSi, StSi, PeDo, and StDo, 5721 of which could be quantified at both the transcriptional and translational levels (Appendix A). The correlation coefficients between the flower organs at the translational level were higher than those at the transcriptional level (Figure 15). A total of 1402 upregulated DEGs, 8971 downregulated DEGs, 227 upregulated DAPs, and 192 downregulated DAPs were screened from PeSi vs. PeDo, of which 45 upregulated and 21 downregulated genes were differentially expressed in both the transcriptome and proteome. A total of 3389 upregulated DEGs, 1942 downregulated DEGs, 249 upregulated DAPs, and 393 downregulated DAPs were screened from StSi vs. StDo, of which 54 upregulated and 91 downregulated genes were differentially expressed in both the transcriptome and proteome (Figure 16).

Additionally, 223 and 277 genes/proteins were expressed in PeSi vs. PeDo and StSi vs. StDo, respectively, and the Pearson correlation coefficients (PCC) between the genes and proteins were 0.324 and 0.492, respectively (Figure 17a). There were 108 and 179 DEGs-DAPs in PeSi vs. PeDo and StSi vs. StDo, respectively, and the PCCs between the related DAPs and DEGs were 0.349 and 0.578, respectively (Figure 17b). These results showed that the correlation between the DEGs and DAPs was higher than that between the total genes and proteins. In the PeSi vs. PeDo and StSi vs. StDo groups, there were 66 and 145 DEGs-DAPs with the same expression trend, respectively, and the PCCs between the DAPs and DEGs were 0.692 and 0.695, respectively (Figure 17c). Among the DEG-DAPs with the same expression trend, 45 DEGs-DAPs with a PCC of 0.671 in PeSi vs. PeDo and 54 DEGs-DAPs with a PCC of 0.855 in StSi vs. StDo were upregulated (Figure 17d). Moreover, 21 DEGs-DAPs with a PCC of 0.735 in PeSi vs. PeDo, and 91 DEGs-DAPs with a PCC of 0.604 in StSi vs. StDo were downregulated (Figure 17e). In PeSi vs. PeDo and StSi vs. StDo, there were 42 and 34 DEGs-DAPs with opposite expression trends, respectively, and their PCCs were −0.191 and −0.081, respectively (Figure 17f).

#### 2.8.2. Analysis of Differentially Expressed Genes/Differentially Abundant Proteins with the Same Expression Trend

GO enrichment analysis was conducted on DEGs-DAPs with the same expression trend (Appendix A), and it was found that, in terms of cell composition, the cytoplasm, cell wall, and mitochondrial membrane were mainly enriched; in terms of molecular function, various enzymes, such as glycosyltransferases, oxidoreductases, hydrolases, and transmembrane transport proteins, were mainly enriched; and in terms of biological process, polysaccharide metabolism and the jasmonic acid signal transduction pathway were mainly enriched. Polysaccharide metabolism genes/proteins include xyloglucan endoglycosyltransferase/hydrolase, endoglucanase, and granule-bound starch synthase, and jasmonic acid signal transduction pathway genes/proteins include jasmonic acid amide synthase and 2-ketoglutarate-dependent dioxygenase (Appendix A). There were four common DEGs-DAPs with the same expression trend, including the non-specific lipid transfer protein (CDL15_Pgr000732) in the transmembrane transport pathway, geraniol 8-hydroxylase (CDL15_Pgr017793) in the secondary metabolic pathway, and the latex protein (CDL15_Pgr011004) and anti-streptomyces griseus albumen inhibitor (CDL15_Pgr008444) in the stress response pathway. In terms of the hormone pathways, one auxin transporter (*LAX2*) and one *JAR1* were found among the DEGs-DAPs with the same expression trend in PeSi vs. PeDo. The expression of *LAX2* in StSi, StDo, and PeDo increased in sequence and was highly expressed in PeDo. The expression of JAR1 in StDo and PeDo was higher than that in StSi and PeSi. One auxin induced in root culture protein 12 (*AIR12*) and one *GH3.17* were found in StSi vs. StDo. Their expression levels were significantly higher in StSi than in StDo, PeDo, and PeSi (Figure 18).

## 3. Discussion

### 3.1. Overview of the Transcriptome and Proteome in Ornamental Pomegranate

In this study, the genes and proteins related to the petaloidy of ornamental pomegranate were screened using combined transcriptome and proteome analyses (Figure 19). A total of 24,567 genes and 5865 proteins were identified, of which 5721 genes were quantified at both the transcriptional and translational levels. The correlation among the flower organs in the proteome was higher than that in the transcriptome, indicating that the differences at the transcriptional level for each flower organ were far greater than the differences at the translational level. In the PeSi vs. PeDo and StSi vs. StDo groups, the correlation between the total genes and total proteins was low, which may be due to the involvement of post-transcriptional regulation, epigenetic modification, post-translational regulation, and other regulatory activities of the messenger RNA (mRNA). Notably, the correlation between the DAPs and DEGs in the two comparison groups was higher than that between the total genes and proteins, indicating that petaloidy affected this correlation. The phenotype also affects the correlation between DEGs and DAPs in plants, such as *Nelumbo nucifera* [18] and *Manihot esculenta* [30]. According to the GO enrichment analysis, cell wall polysaccharide metabolism, jasmonic acid signal transduction, redox balance, and transmembrane transport all had important effects on the petaloidy of ornamental pomegranate in the transcriptome, proteome, and association analyses of the two omics. Previous studies demonstrated that cell wall metabolism affects the mid- to late development of stamens, and genes related to cell wall metabolism regulate the abnormal development of anther walls, leading to stamen abortion [31], as well as affecting petal elongation and morphogenesis [32]. Previous studies have shown that jasmonic acid, auxin, and gibberellin signal transduction pathway genes play important roles in regulating the petaloidy of *Lagerstroemia specious* [20]. A study found that, after spraying exogenous jasmonic acid on the inflorescence of the Chinese cabbage petal degeneration mutant, the petals returned to normal [33]. The redox balance is involved in plant stress resistance and flower development. Previous studies demonstrated that the differential expression of peroxidase isoenzymes may lead to the apetalous trait in *Brassica napus* [34]. Transmembrane proteins can regulate floral organ development by participating in hormone transport. For example, the transmembrane protein *PmARF17* negatively regulates the content of abscisic acid in pistils, stamens, and petals, thereby affecting floral development and the process of pistil abortion in *Prunus mume* [35].

### 3.2. Hormone Pathway Participation in Ornamental Pomegranate Petaloidy

Previous studies have shown that hormone-related genes are involved in the petaloidy of various plants [13,36]. We screened four hormone-related DEGs from the transcriptome (*ARF*s and *GH3.17*), three hormone-related DAPs from the proteome (ILR1, JAR1, and ASR), and four hormone-related DEGs-DAPs from the association analysis of the transcriptome and proteome (*LAX2*, *AIR12*, *GH3.17*, and *JAR1*) (Figure 19). These DEGs/DAPs were involved in the auxin, jasmonate, and abscisic acid pathways. Auxin synthesis, metabolism, transport, and signal transduction play important roles in stamen development and petal growth [36,37]. Previous studies have shown that high concentrations of auxins in the stamen can regulate anther dehiscence, pollen maturation, and filament elongation in *A. thaliana* [38]; promote the elongation of chrysanthemum petal cells; and increase the number of petal basal cells [36]. Indole-3-acetic acid-amido synthetase catalyzes the combination of auxin and amino acid and reduces biological activity, while ILR1 catalyzes the release of free auxin from bound auxin, which increases biological activity [39]. In this study, the expression of ILR1 protein in the PeDo and StDo was higher than that in PeSi and StSi, indicating that ILR1 might promote the petaloidy of ornamental pomegranate. In contrast, the expression of *GH3.17* in the StSi was higher than that in StDo, PeDo, and PeSi, indicating that *GH3.17* might inhibit the petaloidy of ornamental pomegranate. Their expression patterns are consistent with their opposing functions in auxin metabolism. The auxin transport vector is responsible for transporting auxin that is outside the cytoplasm into the cell [40], which is crucial for the early and late development of stamens [41]; the mutation of the auxin inflow vector gene (*AUX1*/*LAX*) leads to structural variation in the inflorescence [42]. The expression of *LAX2* in StSi, StDo, and PeDo increased, and it was the highest in PeDo, indicating that *LAX2* might promote stamen petaloidy and transitional petal development in ornamental pomegranates. Previous studies have found that *ARF* mutants lead to changes in the number of stamens and petals [43], and *ARF17* regulates anther dehiscence by regulating the lignification of the anther endothecium cells [44]. Additionally, we found that the expression of *ARF* increased in StSi, StDo, PeDo, and PeSi, indicating that it might promote ornamental pomegranate petaloidy. However, the expression of the auxin response factor *AIR12* in StSi was higher than that in other flower organs, indicating that auxin signal transduction-related genes have complex regulatory mechanisms.

Jasmonic acid affects the development of flower organs and participates in stress response. It is one of the main hormones involved in late stamen development and petal formation. Jasmonic acid-amido synthetase protein catalyzes the formation of active jasmonate isoleucine conjugates which in turn activates the downstream signal transduction pathway [45]. Xiao et al. found that *JAR1* regulates the opening, closing, and cracking of rice anthers [46]. We found that the expression of *JAR1* in StDo and PeDo was higher than that in StSi and PeSi, indicating that it may promote stamen petaloidy. Nonetheless, in a transcriptome study of *Malus spectabilis*, the expression of *JAR1* in single-petal flowers was higher than that in double-petal flowers, which may be owing to species specificity [19]. Notably, GH3 also functions as a jasmonate amide synthetase that catalyzes jasmonate and reduces several biological activities [47]. In this study, *GH3.17* and *JAR1* regulated the petaloidy of ornamental pomegranate, which was consistent with their functions in hormone metabolism. 

The level of abscisic acid, a stress hormone, rapidly increases under stressful conditions, promoting flower bud differentiation and other developmental processes. Previous studies have found that the expression of abscisic-acid-related genes in single-petal flowers was higher than that in double-petal flowers of *Clematis patens* [48]. Abscisic acid stress ripening-induced protein is a common component of the abscisic acid and stress response pathways [49]. Previous studies have found that the *ASR* and TF genes are involved in flower development; however, their specific functions in flower organogenesis have not been reported [50]. We found that the expression of ASR in StDo and PeDo was lower than that in StSi and PeSi, indicating that it might inhibit ornamental pomegranate petaloidy.

### 3.3. Transcription Factor Participation in Ornamental Pomegranate Petaloidy

In this study, fourteen TF DEGs were selected from the transcriptome, and two were selected from the proteome (Figure 19). Among them, one *LOB*, one *MEF2*, one *MYB*, and two *EREBP* genes were expressed at their highest levels in StDo, indicating that they might promote stamen petaloidy, and one *C3H* and one trihelix gene increased successively in StSi, StDo, and PeDo, indicating that they might promote stamen petaloidy and transitional petal growth. The *EREBP* transcription factor is mainly involved in plant stress response [51], but its effect on floral organ development is still unclear. The *LOB* transcription factor participates in pollen development [52], the *MEF2* transcription factor belongs to the MADS family and participates in flower development [53], and the *MYB* transcription factor regulates stamen and petal formation [54]. Additionally, *C3H* encodes coumarin-3-hydroxylase, which is a key enzyme in the lignin synthesis pathway, and trihelix regulates the morphogenesis of floral organs, such as sepals and petals [55]. Hormones are closely associated with these TFs. For example, jasmonic acid can promote TFs, such as EREBPs and MYBs, through the receptor protein COL1 and receptor protein complex JAZ, thus activating the expression of downstream genes [45].

## 4. Materials and Methods

### 4.1. Plant Materials and Sample Collection

The pomegranate varieties ‘Nanlindanbanhong’, with single-petal flowers, and ‘Nanlinchongbanhong’, with double-petal flowers, were planted in the nursery of Nanjing Forestry University (32°4′38″ N, 118°49′5″ E). The two varieties had the same plant type, branch type, and flower color. The number of petals and petalized stamens in the two varieties was counted, and the flower morphology was observed. StSi (stamens of the single-petal flowers) and PeSi (petals of the single-petal flowers) on pomegranate varieties ‘Nanlindanbanhong’, and StDo (stamens of the double-petal flowers including petalized stamens) and PeDo (petals of the double-petal flowers including transitional petals) on pomegranate varieties ‘Nanlinchongbanhong’, were collected for transcriptome and proteome sequencing, with three biological replicates for each group. All the samples were immediately frozen in liquid nitrogen and kept at −80 °C.

### 4.2. Transcriptome Sequencing and Assembly and Gene Functional Annotation

The RNA integrity was assessed by choosing an RNA integrity number that was greater than 7.0 for all samples. RNA reagent (RNAprep Pure Kit, Centrifugal Column Type, Tiangen, Beijing, China) was used to extract total RNA from the samples. Isolated mRNA was spiked with oligo beads (dT) and fragmented into short fragments. The complementary DNA fragments (cDNA) were then sequenced with a platform BGISEQ-500 (Huada Biotechnology, Wuhan, China). The reads that contained adapters, poly N, or low-quality sequences were excluded. The high-quality reads were mapped to the reference transcriptome of pomegranate ‘Dabenzi’ (DNA DataBank of Japan/European Nucleotide Archive/GenBank, MTKT00000000) using HISAT [56] and the default parameters. RSEM [57] was then used to compute gene and transcript expression levels, and the transcriptome quality was tested based on the randomness, coverage, and saturation distributions. Transcripts from each sample were then reconstructed using StringTie (v1.0.4; http://ccb.jhu.edu/software/stringtie; accessed on 10 January 2020), and Cuffmerge software (v2.2.1; http://cole-trapnell-lab.github.io/cufflinks; accessed on 15 January 2020) was used to integrate the reconstructed information from all the samples. The screened-out new transcripts were then compared to transcripts with reference annotations. Subsequently, CPC (v0.9-r2; http://cpc.cbi.pku.edu.cn; accessed on 18 January 2020) was used to predict the protein-coding potential of the novel transcripts. Additionally, the software getorf (http://emboss.sourceforge.net/apps/cvs/emboss/apps/getorf.htm; EMBOSS:6.5.7.0; accessed on 20 January 2020) was used to detect the open reading frame (ORF) of the unigene program, hmmsearch (v3.0; http://hmmer.org; accessed on 22 January 2020) was used to align ORFs to the protein domain of the transcription factor, and transcription factor capacity was identified based on the Plant Transcription Factor Database.

### 4.3. RNA Sequencing Data Validation Using Quantitative Real-Time Polymerase Chain Reaction

The RNA-seq results were validated using qRT-PCR and the expression of 12 randomly selected genes. The cDNA was obtained by reverse transcription using the extracted total RNA as a template. The reverse transcription reagent for qRT-PCR was the Hiscript II Q RT Super Mix (+gDNA Wiper; Vazyme, Nanjing, China; # R223-01). Then, the designed qRT-PCR primers (Appendix A) were used to carry out the polymerase chain reaction with the fluorescent reagent AceQ qRT-PCR SYBR Green Master Mix (Vazyme; # Q111-02/AA). The reaction conditions for the amplification were: 95 °C for 5 min; 95 °C for 10 s, and 58 °C for 30 s, for 40 cycles; 95 °C for 15 s, 60 °C for 1 min, 95 °C for 15 s; and 40 °C for 5 min. The reaction volume consisted of 5 μL of AceQ qPCR SYBR Green Master Mix, 0.2 μL of primer 1, 0.2 μL of primer 2, 1 μL of 10 × diluted cDNA, and 4.6 μL of distilled water. Data were analyzed using three replicates, and the relative expression was calculated using the 2^−ΔΔCT^ method.

### 4.4. Protein Extraction and Trypsin Digestion

The samples were ground to powder with liquid nitrogen and then added to a four-volume cracking buffer followed by ultrasonic treatment on ice. An equal volume of Tris-Balanced phenol was added to remove residual debris by centrifugation. Five volumes of 0.1 M ammonium acetate/methanol were then added to precipitate the protein overnight. The protein was redissolved in 8 M urea, and the protein concentrations were detected with a BCA kit (PA115, Tiangen, China).

The protein solution was reduced with 5 mM dithiothreitol at 56 °C for 30 min, followed by alkylation with 11 mM iodoacetamide under dark conditions for a further 15 min. The protein sample was then diluted to a urea concentration of no more than 2 M by the addition of 100 mM TRAB. Lastly, trypsin was added at a 1:50 mass ratio of trypsin to protein and incubated for 4 h at a mass ratio of 1:100 trypsin to protein.

### 4.5. Liquid Chromatography with Tandem Mass Spectrometry Analysis and Database Search

The peptides were dissolved in solvent A (0.1% formic acid) and separated using a gradient of solvent B (0.1% formic acid in 98% acetonitrile), 6–23% solvent B for 30 min, 25–35% solvent B for 8 min, and 80% solvent B for 6 min. They were then subjected to segregation using the ultra-high performance liquid phase NanoElute system (EASY-nLC 1000 UPLC, ThermoFisher, Waltham, MA, USA) and injected into the NSI source followed by tandem mass spectrometry (MS/MS) in an Ultra High Resolution Liquid Mass Spectrometer (Q ExactiveTM Plus, ThermoFisher, USA) coupled in-line with the UPLC system. An electrospray voltage of 2.0 kV was used. For the full scan, the m/z scan ranged between 350 and 1800, and the intact peptides were detected using an ion trap analyzer (Orbitrap, ThermoFisher, USA) at a resolution of 70,000 s. Peptides were selected for MS/MS analysis using the NCE setting as 28, and fragments were detected using Orbitrap at a resolution of 17,500 s. A data-dependent procedure that alternated between one MS scan followed by 20 MS/MS scans with 15.0 dynamic exclusion was conducted. The automatic gain control was set to 5E4. The first fixed mass was set to 100 m/z, while the scanning range of the secondary mass spectrometry was set to 100–1700 m/z. The parallel cumulative serial fragmentation mode was used for data acquisition.

The secondary mass spectrometry data were searched using MaxQuant (v1.6.6.0; http://www.maxquant.org/; accessed on 17 September 2020). The database that was used was the Punica_granatum_22663_NCBI_20190902 (29,127 sequences; the date of access: 14 July 2020), which included an inverse database to compute the false positive rate that was caused by random matching as well as a common pollution database. Trypsin/P was specified as a cleavage enzyme allowing up to 4 missing cleavages. The mass tolerance for precursor ions was set to 20 ppm in the first search and 5 ppm in the main search, while the mass tolerance for fragment ions was set to 0.02 Da. Carbamidomethyl on Cys was specified as a fixed modification, and acetylation modification and oxidation on Met were specified as variable modifications. The false discovery rate was adjusted to <1% and the minimum score for modified peptides was set to >40. The peptide length and protein molecular weight distributions were determined to assess the quality of the proteome sequencing.

### 4.6. Bioinformatic Analysis of the Transcriptome and Proteome

The fragments per kilobase of transcript per million mapped reads method and the RSEM software package (http://deweylab.biostat.wisc.edu/rsem/rsem-calculate-expression.html; accessed on 3 July 2022) were used to compute the gene expression levels in the transcriptome, and the label-free quantitation intensity was used to calculate the relative quantitative value of the proteins in the proteome. The detection of the DEGs and DAPs was based on the Poisson distribution [58]. The genes/proteins with a log2 fold change > 1 and a Q-value ≤ 0.001 were selected as the DEGs/DAPs. Online software (https://www.omicshare.com/tools; accessed on 15 July 2022) was used to conduct the GO and KEGG classification and enrichment analyses.

## 5. Conclusions

The aim of this study was to perform transcriptomic and proteomic sequencing of the stamens and petals in single-petal flowers and double-petal flowers of ornamental pomegranates during the flowering period. A total of 24,567 genes and 5865 proteins were identified, of which 5721 genes were quantified at both transcriptional and translational levels. In the petal and stamen comparison groups, the association between the DAPs and DEGs was higher than that between the total genes and proteins, indicating that petaloidy has an impact on the correlation between the genes and proteins. The enrichment results of the transcriptome, proteome, and correlation analyses showed that cell wall metabolism, jasmonic acid signal transduction, redox balance, and transmembrane transport had important effects on petaloidy. Nine hormone-related DEGs/DAPs were selected, among which ARF, ILR1, LAX2, and JAR1 may promote petal doubling. Sixteen transcription factor DEGs/DAPs were selected, among which EREBP, LOB, MEF2, MYB, C3H, and trihelix may promote petal doubling. This study provides transcriptomic and proteomic data on the petaloidy molecular mechanism and provides a theoretical foundation for double-flower breeding in ornamental pomegranate. Further functional verification of hormone-related and TF genes could offer more insights and allow for an in-depth understanding of transcriptional and translational involvement in regulating flower organ development. In this study, transcriptome and proteomic data were obtained for the breeding of new double-petal flower varieties of ornamental pomegranate; however, functional verification for hormone-related genes and TF genes has not been carried out. In particular, the relationship between these candidate genes and ABCDE model genes warrants further investigation.

## Figures and Tables

**Figure 1 plants-12-02402-f001:**
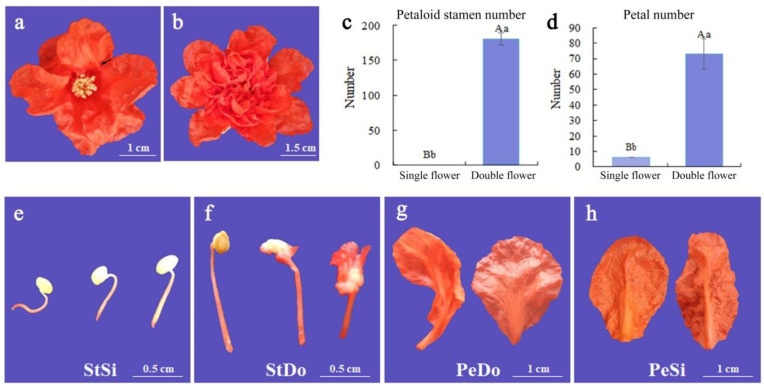
Phenotypic characteristics and morphological indicators of ornamental pomegranate flowers. (**a**) Morphology of single-petal flower; (**b**) morphology of double-petal flower; (**c**) petaloid stamen number of single and double-petal flower; (**d**) petal number of single and double-petal flowers; (**e**) stamen morphology of single-petal flower; (**f**) stamen morphology of double-petal flower, including petaloid stamens; (**g**) petal morphology of double-petal flower, in which the left side is transitional petal, and the right side is normal petal; (**h**) morphology of single petals. Lowercase letters represent the difference is significant (*p* < 0.05), uppercase letters represent the difference is highly sifnificant (*p* < 0.01).

**Figure 2 plants-12-02402-f002:**
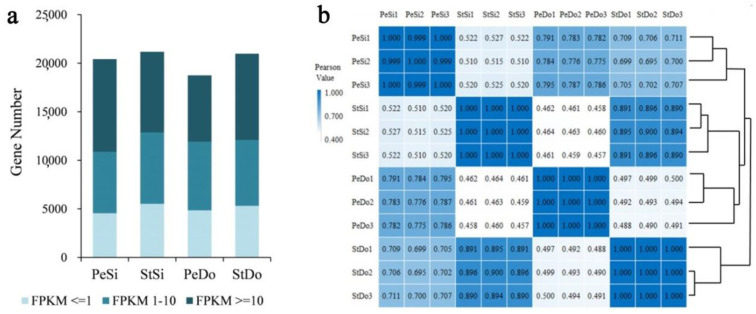
Basic analysis of transcriptome sequencing in ornamental pomegranate. (**a**) Expression accumulation diagram; (**b**) Pearson correlation coefficient heatmap.

**Figure 3 plants-12-02402-f003:**
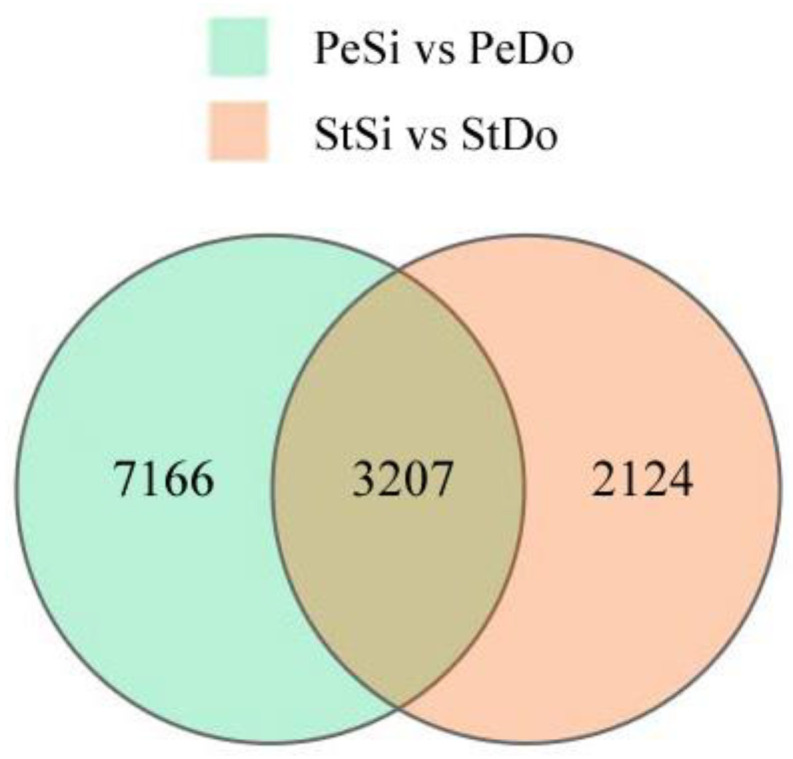
Venn diagram of petaloidy-related DEGs.

**Figure 4 plants-12-02402-f004:**
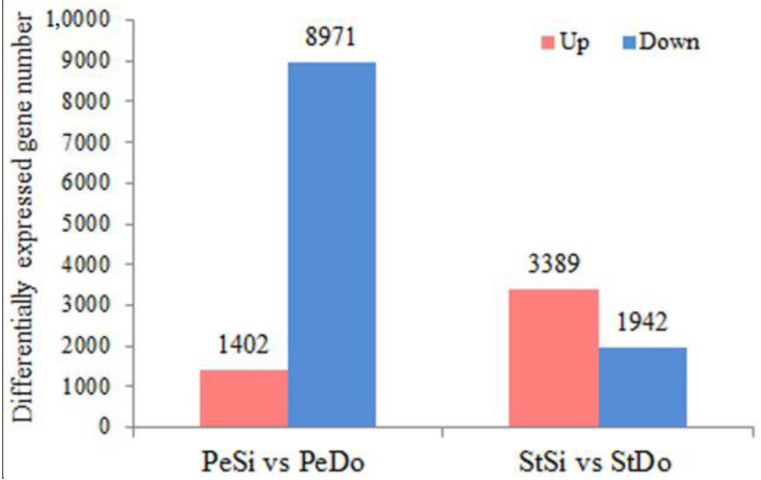
Bar graph of petaloidy-related DEGs.

**Figure 5 plants-12-02402-f005:**
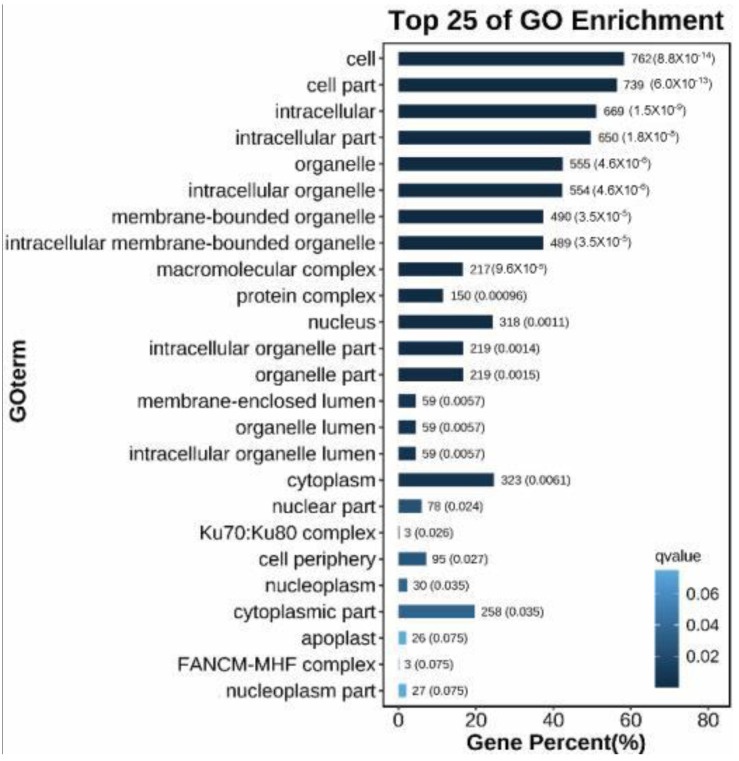
Enrichment map of GO cell components of common DEGs.

**Figure 6 plants-12-02402-f006:**
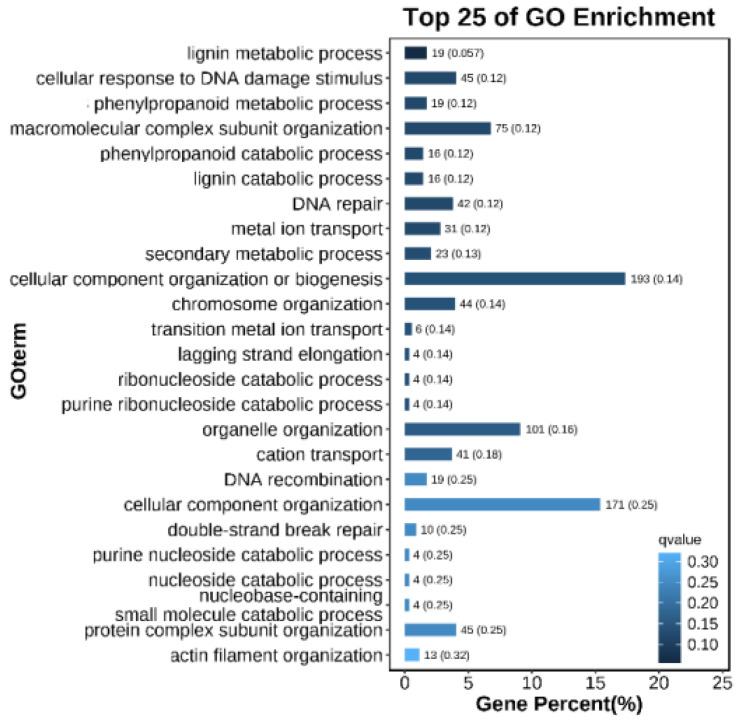
Enrichment map of GO biological process of common DEGs.

**Figure 7 plants-12-02402-f007:**
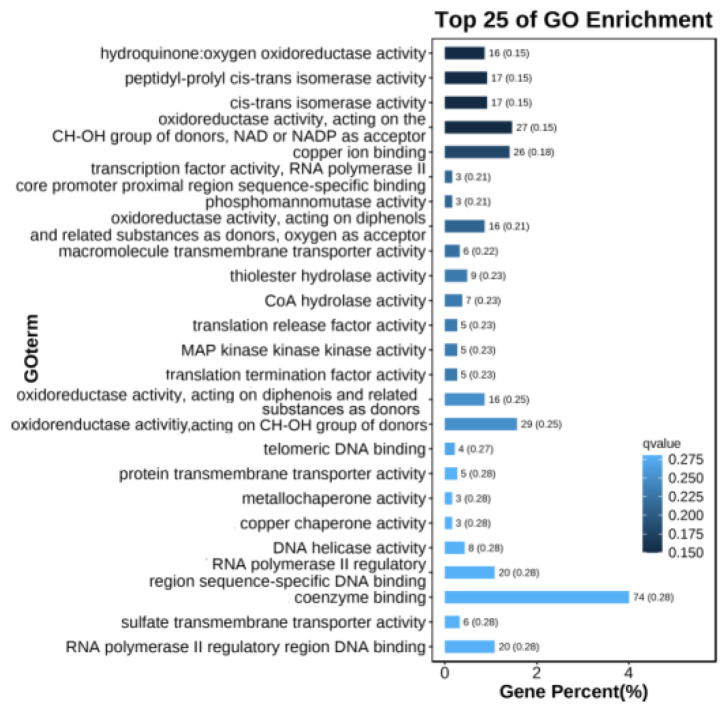
Enrichment map of GO molecular function of common DEGs.

**Figure 8 plants-12-02402-f008:**
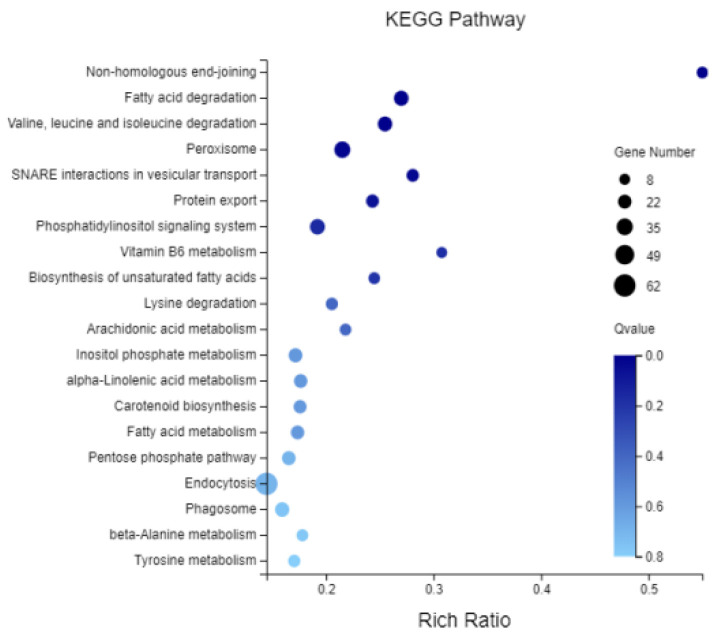
Enrichment map of KEGG pathway of common DEGs.

**Figure 9 plants-12-02402-f009:**
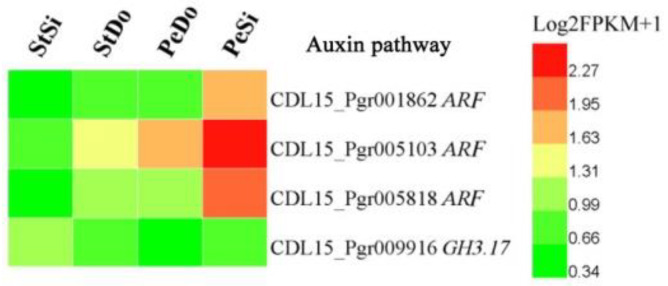
Heatmap of expression pattern for hormone-related DEGs.

**Figure 10 plants-12-02402-f010:**
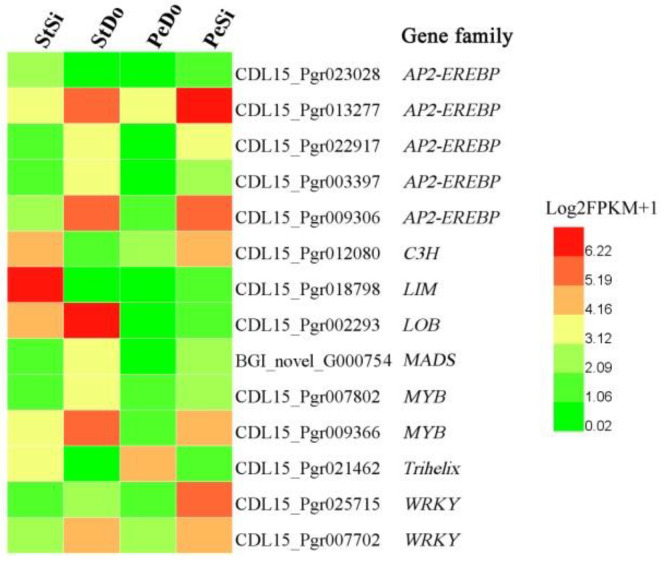
Heatmap of expression pattern for transcription factor DEGs.

**Figure 11 plants-12-02402-f011:**
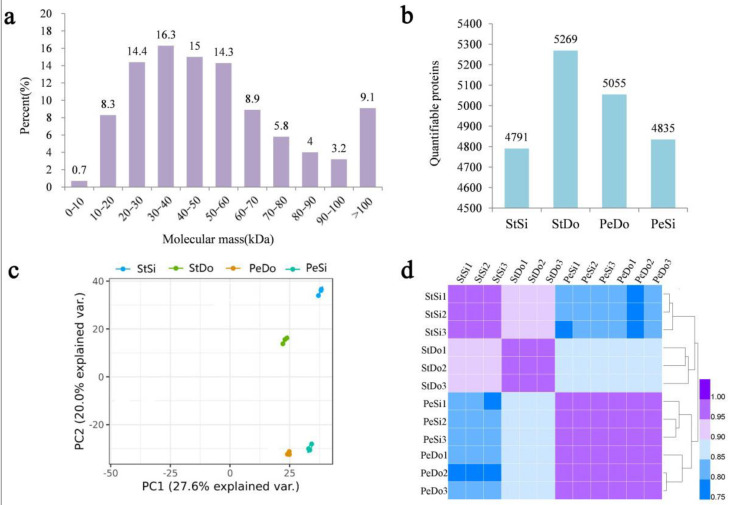
Basic analysis of proteome sequencing in ornamental pomegranate. (**a**) Molecular mass distribution of identified proteins; (**b**) number of quantifiable proteins; (**c**) scatter diagram of principal component analysis t; (**d**) heatmap of Pearson correlation coefficient.

**Figure 12 plants-12-02402-f012:**
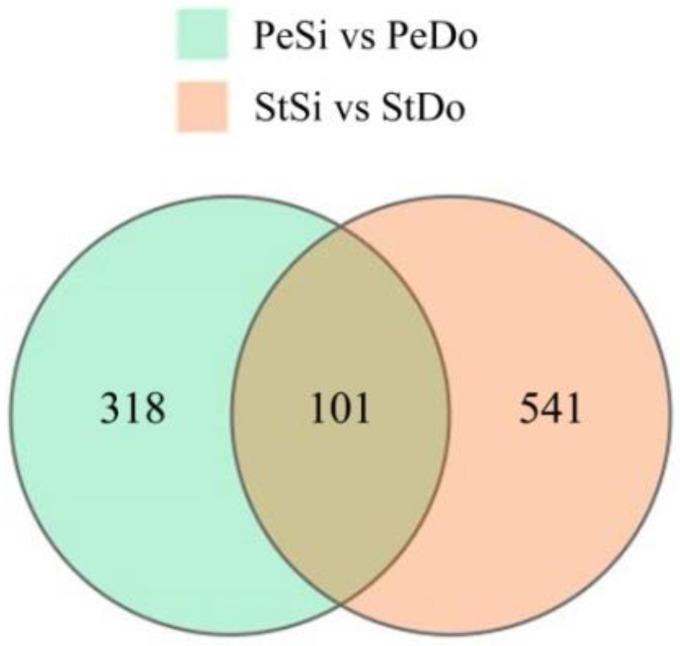
Venn diagram of petaloidy-related DAPs.

**Figure 13 plants-12-02402-f013:**
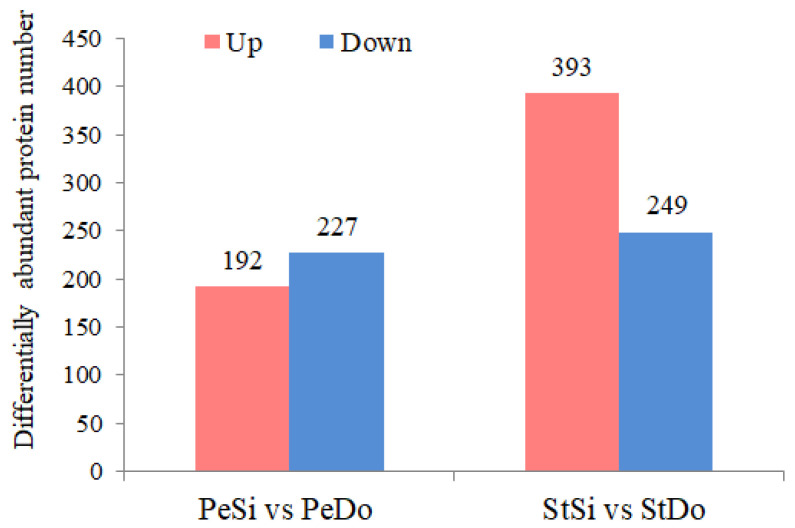
Bar graph of petaloidy-related DAPs.

**Figure 14 plants-12-02402-f014:**
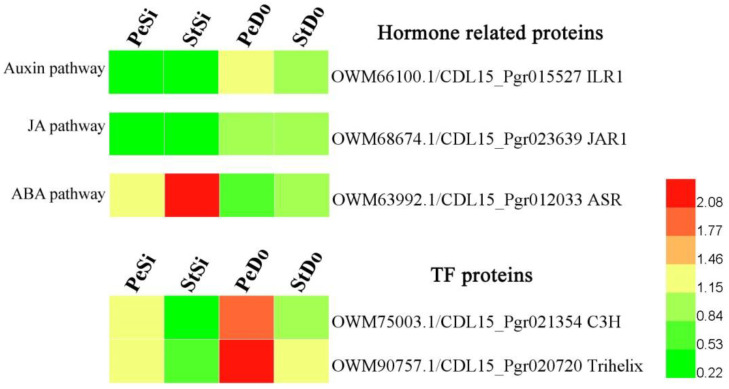
Heatmap of expression pattern for petaloidy-related proteins.

**Figure 15 plants-12-02402-f015:**
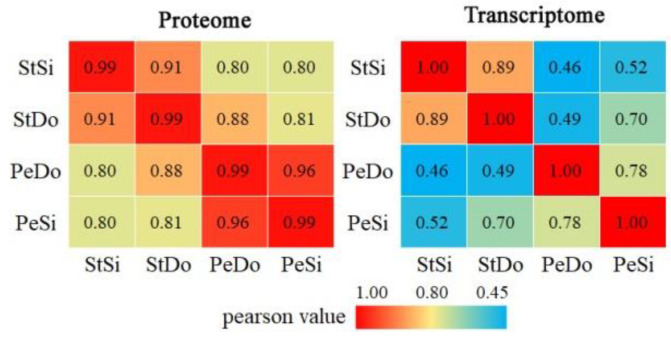
Heatmap of Pearson correlation coefficient between transcriptome and proteome.

**Figure 16 plants-12-02402-f016:**
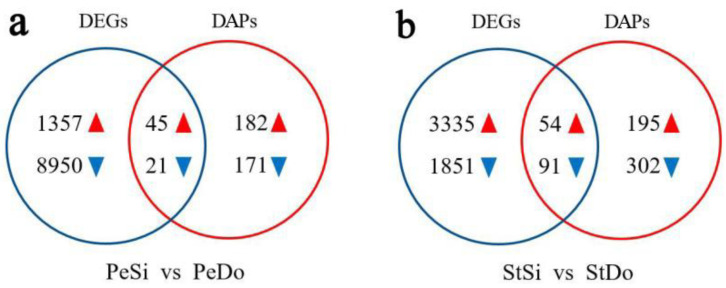
Venn diagram of DEGs and DAPs. Note: Venn diagram of DEGs and DAPs in (**a**) PeSi vs. PeDo and (**b**) StSi vs. StDo.

**Figure 17 plants-12-02402-f017:**
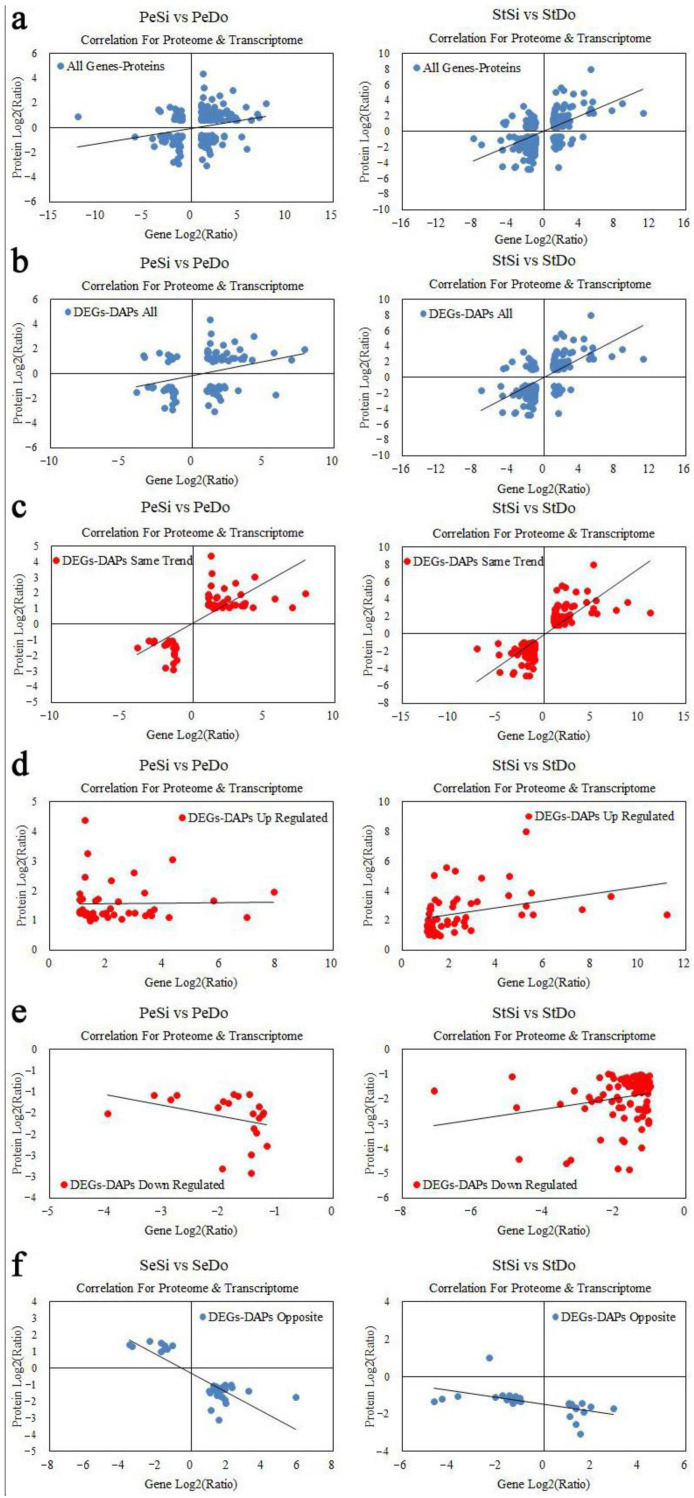
Relationship between transcriptomic data and proteomic data. (**a**) Scatterplots of relationship between genes identified in both transcriptomic data and proteomic data; (**b**) scatterplots of relationship between DEGs and DAPs; (**c**) scatterplots of relationship between DEGs and DAPs with similar changing trend; (**d**) scatterplots of relationship between DEGs and DAPs that were upregulated; (**e**) scatterplots of relationship between DEGs and DAPs that were downregulated; (**f**) scatterplots of relationship between DEGs and DAPs with opposite changing trend.

**Figure 18 plants-12-02402-f018:**
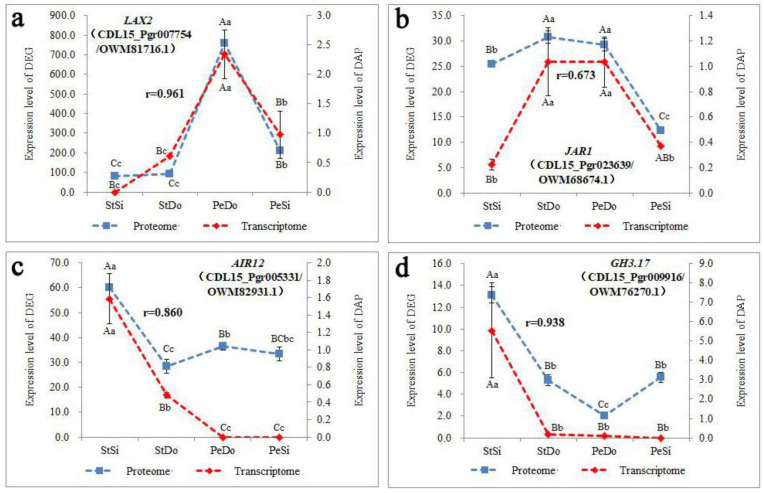
Expression patterns of *LAX2*, *JAR1*, *AIR12*, and *GH3.17* in transcriptome and proteome. (**a**) Expression pattern of *LAX2* in transcriptome and proteome; (**b**) expression pattern of *JAR1* in transcriptome and proteome; (**c**) expression pattern of *AIR12* in transcriptome and proteome; (**d**) expression pattern of *GH3.17* in transcriptome and proteome. Lowercase letters represent the difference is significant (*p* < 0.05), uppercase letters represent the difference is highly sifnificant (*p* < 0.01).

**Figure 19 plants-12-02402-f019:**
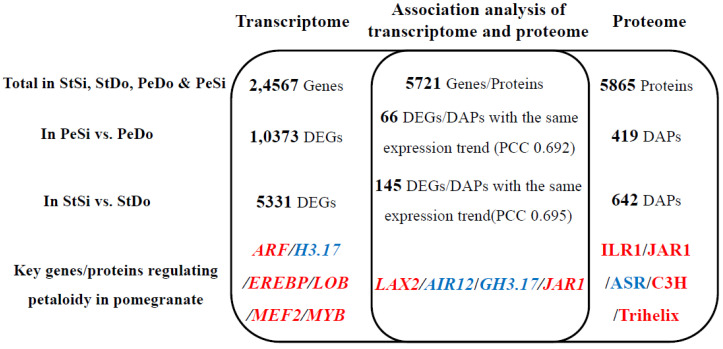
Overview of transcriptome, proteome, and correlation analysis between two omics. Genes/proteins promoting petaloidy are shown in red, while those inhibiting petaloidy reversely are shown in blue.

**Table 1 plants-12-02402-t001:** Summary statistics of sequencing data in the transcriptome of ornamental pomegranate.

Samples	Raw Reads (M)	Total Clean Reads (M)	Total Clean Bases (Gb)	Clean Bases of Q20 (%)	Mapped Reads
PeSi1	47.33	43.59	6.54	96.34	40.63 (93.22%)
PeSi2	49.08	43.74	6.56	97.08	40.85 (93.39%)
PeSi3	45.57	42.04	6.31	96.89	39.67 (94.37%)
StSi1	47.33	43.19	6.48	96.72	40.34 (93.39%)
StSi2	47.33	43.30	6.50	96.86	40.55 (93.65%)
StSi3	49.08	43.98	6.60	96.16	40.30 (91.63%)
PeDo1	49.08	43.53	6.53	96.75	40.62 (93.31%)
PeDo2	49.08	43.38	6.51	96.88	40.41 (93.15%)
PeDo3	50.83	44.70	6.70	96.63	41.56 (92.98%)
StDo1	49.08	44.07	6.61	97.06	40.66 (92.27%)
StDo2	47.33	43.10	6.47	97.03	39.99 (92.79%)
StDo3	47.33	42.50	6.38	97.03	39.23 (92.31%)

**Table 2 plants-12-02402-t002:** Classification of petaloidy-related DEGs of ornamental pomegranate.

Function Category	Gene Name (Gene ID)
Signal transduction	serine/threonine-protein phosphatase (CDL15_Pgr004407)
	mitogen-activated protein kinase (CDL15_Pgr001413, CDL15_Pgr003484, CDL15_Pgr019927, CDL15_Pgr006403)
	interleukin-1 receptor-associated kinase 4 (CDL15_Pgr008721, CDL15_Pgr012996, CDL15_Pgr014246)
	1-phosphatidylinositol-4-phosphate 5-kinase (CDL15_Pgr003539, CDL15_Pgr019098, CDL15_Pgr026956, CDL15_Pgr013697, CDL15_Pgr018872)
DNA repair	centromere protein S (CDL15_Pgr011715, CDL15_Pgr015888, CDL15_Pgr023953)
	ATP-dependent DNA helicase 2 (CDL15_Pgr007504, CDL15_Pgr025038, CDL15_Pgr028666)
	RuvB-like protein 1 (CDL15_Pgr006382)
	DNA replication licensing factor (CDL15_Pgr024576, CDL15_Pgr007997)
	DNA repair protein (CDL15_Pgr011609)
Translation release	peptide chain release factor 1 (CDL15_Pgr018587, CDL15_Pgr021630)
	peptide chain release factor 2 (CDL15_Pgr003092, CDL15_Pgr018186)
	peptidyl-tRNA hydrolase (CDL15_Pgr003817)
Transmembrane transport	protein transport protein (BGI_novel_G000167)
	mitochondrial import inner membrane translocase subunit (CDL15_Pgr012086, CDL15_Pgr026447)
Redox balance	peroxidase (CDL15_Pgr017144)
	laccase (CDL15_Pgr000587, CDL15_Pgr006286, CDL15_Pgr006287, CDL15_Pgr007746, CDL15_Pgr007747, CDL15_Pgr007840)
Lignin metabolism	cinnamyl-alcohol dehydrogenase (CDL15_Pgr003366)
	caffeic acid 3-O-methyltransferase (CDL15_Pgr023102)
	laccase (CDL15_Pgr000587, CDL15_Pgr006286, CDL15_Pgr006287, CDL15_Pgr007746, CDL15_Pgr007747, CDL15_Pgr007840)

## Data Availability

The data presented in this study are available in the text and Appendix A.

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
