# Peer review of "Combined Transcriptome and Proteome Analysis Provides Insights into Petaloidy in Pomegranate"

_plants, 2023, doi:10.3390/plants12132402_

Round 1

Reviewer 1 Report

The manuscript by Huo and colleagues tries to investigate the molecular mechanism of petaloidy in ornamental pomegranates. To this end, the authors used RNA sequencing (RNA-Seq) and LC-MS/MS to monitor a wide range of transcripts and proteins. Subsequent bioinformatics analyses identified several DEGs and DEPs and characterized them by functional enrichment analyses. The RNA-Seq data were validated by qRT-PCR. I had some concerns about the method and the reproducibility.

Major concerns:

1) The authors conducted combined transcriptome and metabolome analyses, however, true integration of them and knowledge extraction were failed.

2) Many figures

There were many figures in the manuscript. The authors should integrate the relevant figures into single figure.

3) Data availability

The RNA-Seq and proteome data are very useful for us. The authors should deposit your RNA sequencing data by an Illumina NovaSeq platform in public repositories (e.g., NCBI SRA, https://www.ncbi.nlm.nih.gov/sra). Also, could you please open your proteome data in community-approved data repository?

Other minor comments:

Figures and Tables

1) Fig. 1c and 1d: What are the bar graphs and error bars? Average? Median?

2) Fig. 8: What is the x-axis?

3) Fig. 18: What are the error bars?

See above.

Reviewer 2 Report

The aim of this study was to perform transcriptomic and proteomic sequencing of the stamens and petals in single-petal flowers and double-petal flowers of ornamental pomegranates during the flowering period. The enrichment results of the transcriptome, proteome, and correlation analyses showed that cell wall metabolism, jasmonic acid signal transduction, redox balance, and transmembrane transport had important effects on petaloidy. This study provides transcriptomic and proteomic data on the petaloidy molecular mechanism and provides a theoretical foundation for double-flower breeding in ornamental pomegranate. Further functional verification of hormone-related and TF genes could offer more insights and allow for an in-depth understanding of transcriptional and translational involvement in regulating flower organ development. Transcriptome and proteomic data were obtained for the breeding of new double-petal flower varieties of ornamental pomegranate; however, functional verification for hormone-related genes and TF genes have not been carried out.

Consider a few of comments in the attached file.

English language is fine; missing only two dots for "et al.", for two references,  [18], [29], page 2
